# Exploratory Study Analyzing the Urinary Peptidome of T2DM Patients Suggests Changes in ECM but Also Inflammatory and Metabolic Pathways Following GLP-1R Agonist Treatment

**DOI:** 10.3390/ijms241713540

**Published:** 2023-08-31

**Authors:** Sonnal Lohia, Justyna Siwy, Emmanouil Mavrogeorgis, Susanne Eder, Stefanie Thöni, Gert Mayer, Harald Mischak, Antonia Vlahou, Vera Jankowski

**Affiliations:** 1Center of Systems Biology, Biomedical Research Foundation of the Academy of Athens, 11527 Athens, Greece; 2Institute for Molecular Cardiovascular Research, RWTH Aachen University Hospital, 52074 Aachen, Germany; 3Mosaiques Diagnostics GmbH, 30659 Hannover, Germany; 4Department of Internal Medicine IV (Nephrology and Hypertension), Medical University Innsbruck, 6020 Innsbruck, Austriagert.mayer@i-med.ac.at (G.M.)

**Keywords:** T2DM, GLP-1R agonists, CE-MS, urine biomarker, peptidomics, bioinformatics, *COL1A1*, *COL3A1*, insulin resistance, inflammation

## Abstract

Type II diabetes mellitus (T2DM) accounts for approximately 90% of all diabetes mellitus cases in the world. Glucagon-like peptide-1 receptor (GLP-1R) agonists have established an increased capability to target directly or indirectly six core defects associated with T2DM, while the underlying molecular mechanisms of these pharmacological effects are not fully known. This exploratory study was conducted to analyze the effect of treatment with GLP-1R agonists on the urinary peptidome of T2DM patients. Urine samples of thirty-two T2DM patients from the PROVALID study (“A Prospective Cohort Study in Patients with T2DM for Validation of Biomarkers”) collected pre- and post-treatment with GLP-1R agonist drugs were analyzed by CE-MS. In total, 70 urinary peptides were significantly affected by GLP-1R agonist treatment, generated from 26 different proteins. The downregulation of MMP proteases, based on the concordant downregulation of urinary collagen peptides, was highlighted. Treatment also resulted in the downregulation of peptides from *SERPINA1*, *APOC3*, *CD99*, *CPSF6*, *CRNN*, *SERPINA6*, *HBA2*, *MB*, *VGF*, *PIGR*, and *TTR*, many of which were previously found to be associated with increased insulin resistance and inflammation. The findings indicate potential molecular mechanisms of GLP-1R agonists in the context of the management of T2DM and the prevention or delaying of the progression of its associated diseases.

## 1. Introduction

In 2021, the International Diabetes Federation (IDF) predicted a five-fold increase in the worldwide adult population of diabetes mellitus, from 150 million in 2000 [1] to 783 million in 2045 [2], of which type II diabetes mellitus (T2DM) is said to account for approximately 90% of all cases [3]. T2DM is characterized as a chronic disease [4] and is diagnosed with increased levels of blood glucose, or hyperglycemia [5]. T2DM has been associated with macrovascular complications, such as atherosclerosis, myocardial infarction, and diabetic foot syndrome, as well as microvascular complications, such as neuropathy, retinopathy, and nephropathy [6,7,8]. Since no cure exists, efforts to control and treat T2DM have escalated [9]. The approach of lowering glucose plasma levels remains the most sought-after treatment and has recently witnessed novel and effective advances. These, along with metformin, commonly include treatment with glucagon-like peptide-1 receptor (GLP-1R) agonists.

The GLP-1R agonists have dramatically transformed patient care guidelines for T2DM [10]. In comparison to the other antihyperglycemic medications, GLP-1R agonists have established an increased capability to target directly or indirectly six out of the eight core defects (the ‘ominous octet’) associated with T2DM [11,12,13]. In response to food intake, the intestinal L-cell-dependent secretion of GLP-1 is significantly impaired in T2DM patients, which affects the pancreatic β-cell-dependent insulin secretion, eventually resulting in hyperglycemia [14,15,16]. The synthetically produced GLP-1R agonists stimulate GLP-1R in a similar fashion to native GLP-1. Treatment benefits of peptide-based GLP-1R agonists include glycemic control, weight loss, delayed onset of macroalbuminuria, prevention of clinical hyperglycemic episodes and cardiovascular events, and reduced estimated glomerular filtration rate (eGFR) decline, which in turn indirectly result in the inhibition of glucagon secretion and increased secretion of insulin with minimal hypoglycemic risks [8,17,18,19]. Since investigation of the pharmacological effects of GLP-1R agonists has proven to be of great advantage, the focus of research can thus shift to understanding the underlying molecular mechanisms.

Recent advances in the analysis of the urinary peptidome have paved the way for the detection of its alterations in chronic diseases [20]. Urine-based omics studies have especially garnered relevance due to the ease of sample collection, the longer stability of the peptides, and the possibility of larger cohort studies thanks to the non-invasive collection of urine. In addition, the water-soluble and charged nature of urinary peptides enable their uncomplicated mass spectrometric (MS)-based detection [21]. The urinary peptidomic analysis in this study was performed by a capillary electrophoresis coupled to mass spectrometry (CE-MS) technique, which supports the identification of naturally occurring urinary peptides and peptidomic changes in response to drug interventions [22]. Specifically, CE-MS possesses the potential to combine a highly efficient separation technology with MS to detect and identify peptides. In the context of T2DM, CE-MS has successfully identified low-molecular-weight urinary peptides with the potential to act as biomarkers for disease diagnosis and prognosis [23]. For instance, the multi-dimensional classifier CKD273 has been established to predict the development of diabetic kidney disease in T2DM patients [24]. Recently, our research group successfully investigated the beneficial effects of Irbesartan, a renoprotective drug, on the urinary peptidome of T2DM patients by applying CE-MS technology [25].

In this exploratory study, we aimed to assess the effect of treatment with GLP-1R agonists on the urinary peptidome of T2DM patients in an untargeted peptidomics approach, aiming to ultimately shed some light on the underlying molecular mechanisms defining response.

## 2. Results

### 2.1. Study Design

The exploratory and untargeted urinary peptidomic approach in this study was conducted to analyze the effect of GLP-1R agonist treatment on the urinary peptidome of T2DM patients, as illustrated in Figure 1. Briefly, urine samples from thirty-two T2DM patients collected at two time points: pre-treatment and post-treatment, with a time interval of 11.5 ± 3.72 months in between, were analyzed. Intervention with a GLP-1R agonist was introduced at 4.4 ± 4.11 months from the pre-treatment visit, and the treatment duration until the post-treatment visit was 6.9 ± 3.59 months.

### 2.2. Clinical Information

The clinical information of the thirty-two T2DM patients participating in this study from visits pre- and post-GLP-1R agonist treatment is provided in Table 1. The mean age for the patients at the first sample collection was 63.7 ± 7.25 years, and 56.3% of them were females. As depicted in Table 1, no statistically significant difference between the clinical levels of hemoglobin A1C (HbA1C), body weight, body mass index (BMI), systolic and diastolic blood pressure (SBP/DBP), eGFR, albuminuria to creatinine ratio, and urinary creatinine was observed after the treatment with GLP-1R agonists.

### 2.3. Peptidomic Analysis

Urinary peptidomic analysis resulted in a list of 329 sequenced peptides that passed the thresholds of frequency (i.e., being detected in at least 10 out of 32 samples or with at least 30% frequency in one group) and area under the receiver operating curve (AUC ≥ 0.60, as described in the methods section). Further statistical assessment, including Benjamini-Hochberg FDR adjustment, yielded a list of 70 (out of the 329 sequenced peptides) statistically significant peptides (adjusted *p*-value < 0.05) affected by the GLP-1R agonist treatment (detailed information on the peptide identifications is provided in Appendix A). The distribution of peptide intensity of these 70 statistically significant peptides amongst the 329 peptides was also examined, indicating that they span the whole detected intensity range (Figure 2a—red spots). Volcano plot analysis highlighted that the majority (66 out of 70 peptides) of the statistically significant peptides were detected at decreased abundance in the post- versus pre-treatment samples (Figure 2b), which may also be seen in the respective CE-MS spectra (Figure 2c,d).

The 70 statistically significant peptides were identified as fragments of 26 proteins, as shown in Table 2. Most of the peptides (59 out of 70) originated from the collagen family of proteins, and 41 out of the 59 collagen peptides belonged to three collagen proteins, namely, collagen alpha-1(III) chain (P02461; *COL3A1*; *n* = 16), collagen alpha-1(I) chain (P0252; *COL1A1*; *n* = 15), and collagen alpha-2(I) chain (P08123; *COL1A2*; *n* = 10). The remaining identified urinary peptides originated from different proteins (1 peptide each), including alpha-1-antitrypsin (P01009; *SERPINA1*), apolipoprotein C-III (P02656; *APOC3*), CD99 antigen (P14209; *CD99*), cleavage and polyadenylation specificity factor subunit 6 (Q16630; *CPSF6*), Cornulin (Q9UBG3; *CRNN*), corticosteroid-binding globulin (P08185; *SERPINA6*), hemoglobin subunit alpha (P69905; *HBA1*; *HBA2*), myoglobin (P02144; *MB*), neurosecretory protein VGF (O15240; *VGF*), polymeric immunoglobulin receptor (P01833; *PIGR*), and transthyretin (P02766; *TTR*). Four out of the 70 urinary peptides showing a statistically significant upregulation following treatment with GLP-1R agonists in T2DM patients originated from *COL3A1* (*n* = 3) and *COL1A2* (*n* = 1) (Table 2).

Since several peptides from *COL1A1* and *COL3A1* were found to be significantly associated with treatment (Figure 3a,b, respectively), we investigated the alignment of these peptides in their protein structures. For both proteins, peptides appeared evenly distributed, and no specific hot spot in the protein sequence became apparent (Figure 3c,d).

### 2.4. Bioinformatic Analysis

To uncover plausible molecular mechanisms responsible for the observed impact of GLP-1R agonist treatment on urinary peptides in T2DM patients, the proteases potentially responsible for the cleavage of the 70 statistically significant peptides were investigated using Proteasix. In total, 10 endopeptidases were retrieved as a result of the default search with the “Observed Prediction tool” of Proteasix (http://www.proteasix.org, accessed on 6 March 2023), putatively responsible for cleaving 38 urinary peptides (36 downregulated and 2 upregulated) out of the 70 significant peptides. The results are provided in Appendix A. Most of the predicted endopeptidases belonged to the matrix metalloproteinase (MMP) family of proteases (7 out of 10 proteases), responsible for cleaving peptides at both the N’ and C’ terminals. Further proteases predicted to be potentially responsible for cleaving at the N’ terminus belonged to the cathepsin family (*CTSL* and *CTSD*), while those predicted to cleave at the C’ terminus included A disintegrin and metalloproteinase with thrombospondin motifs 5 (*ADAMTS5*). Notably, the proteases *MMP2*, *MMP9*, and *MMP13* were mapped to at least six cleavage sites each.

To further link these findings on a molecular level, a protein-protein interactome was constructed using the 26 parental proteins identified from the 70 urinary GLP-1R agonist-associated peptides. The network consisted of 27 nodes and 113 edges, as depicted in Figure 4, with an enrichment score of significant *p*-value (<1.0 × 10^–16^). While most of the collagen proteins can be observed to interact with all other collagen proteins, interestingly, none of them interact with the non-collagen proteins. Within this protein network, a total of 9 KEGG pathways were predicted to be significantly enriched, including pathways related to protein digestion and absorption, ECM-receptor interaction, the AGE-RAGE signaling pathway in diabetic complications, relaxin signaling, the PI3K-Akt signaling pathway, and platelet activation (detailed results are provided in Appendix A).

## 3. Discussion

In the last decade, GLP-1R agonists have been the recommended and preferred second-line treatment for T2DM patients. Despite the various advantages of GLP-1R agonists over other anti-hyperglycemic drugs, the underlying molecular mechanisms of treatment with GLP-1R agonists have not been studied in depth. Aiming to understand the effect of GLP-1R agonist treatment on T2DM patients, the urinary peptidome of 32 T2DM patients was analyzed with CE-MS. The untargeted peptidomic analysis coupled with statistical tools identified 70 statistically significant (adjusted for multiple testing) differentially abundant urinary peptides between the pre- and post-treatment samples. These urinary peptides originate from 26 parental proteins. The uniform distribution of the intensity of the statistically significant peptides (red spots in Figure 2a) suggested that their observed significant change with GLP-1R agonist treatment was not a function of their abundance in the urine samples. Interestingly, the vast majority of the differentially abundant peptides (66/70) decreased in abundance following GLP-1R treatment.

In total, 59 out of the 70 statistically significant urinary peptides originated from three collagen proteins: *COL3A1* (*n* = 16), *COL1A1* (*n* = 15), and *COL1A2* (*n* = 10). Recently, He et al. [26] reported that a high abundance of collagen peptides is observed in urine samples as a result of proline hydroxylation, which plausibly inhibits their reabsorption in the kidney. Fifty-five out of the 59 collagen peptides were observed to be significantly downregulated following the GLP-1R agonist treatment, while only four peptides (*COL3A1*; *n* = 3 and *COL1A2*; *n* = 1) showed a significant upregulation upon treatment, which may be attributed to variations in the peptide hydroxylation of proline residues and/or varied proteolytic cleavage. In line with the extensive recent report by Mavrogeorgis et al. [27], all the urinary collagen peptides identified in this study were devoid of the signal peptide, N-terminal, and C-terminal pro-peptides, corresponding only to the mature protein region (Figure 3c,d). The observed downregulation of the collagen peptides in our study could therefore represent attenuated degradation of the mature collagen due to increased resistance to proteolytic cleavage or protease inhibition, as earlier suggested [23,28]. However, the possibility of decreased collagen synthesis as a result of treatment cannot also be ruled out.

To further corroborate the above speculation, endopeptidases responsible for putatively producing the 38 statistically significant collagen peptides (no protease activity predicted for *n* = (59−38) = 21 collagen peptides) corresponded primarily to MMPs (89% of the cleavage sites), including MMP2 (21.1%), MMP9 (21.1%), and MMP13 (15.5%). The suggested downregulation in the activity of MMP peptidases, as predicted based on the observed decrease in the abundance of collagen peptides following GLP-1R treatment, is in line with the existing literature. Down-regulation in the expression of MMP9 was observed on treatment with Liraglutide (GLP-1R agonist type) in a study that included induced-DM rabbit models [29]. In another study, 45% and 60% reductions in the activity of MMP2 and MMP9, respectively, in addition to 60% reduced COL1A1 levels, were observed in male C57BL/6 mice on treatment with Semaglutide (GLP-1R agonist type) [30]. Research groups exploring the effect of Exenatide (GLP-1R agonist type) treatment on human coronary artery smooth muscle cells [31] and human retinal pigment epithelium cells [32] reported decreased expression of MMP2 and MMP9, respectively, upon drug treatment. Another study analyzing atherosclerosis-associated biomarkers in T2DM females reported a decrease in MMP2 and MMP9 levels with an increase in GLP-1 and GLP-1R levels [33]. Interestingly, treatment of Human SW1353 with Dulaglutide (GLP-1R agonist type) [34] and Fibroblast-like synoviocytes cell lines with Exenatide [35] also resulted in the downregulation of MMP13 and ADAMTS5 proteases.

On the other hand, collectively, all the non-collagen peptides (11 out of 70 originating from *SERPINA1*, *APOC3*, *CD99*, *CPSF6*, *CRNN*, *SERPINA6*, *HBA2*, *MB*, *VGF*, *PIGR*, and *TTR*) also showed lower abundance after treatment with GLP-1R agonists. In studies analyzing the effect of Liraglutide on T2DM patients, Rafiullah et al. [36] reported the downregulation of the urinary protein *SERPINA1*, and Adiels et al. [37] reported decreased secretion of *APOC3* with treatment. Increased *SERPINA1* levels in serum have been reported in inflammatory diseases including acute kidney injury, glomerulonephritis, minimal change disease, and focal segmental glomerulosclerosis [38]. Increased *APOC3* levels in plasma have been reported in diabetic kidney disease patients and are increasingly considered an independent risk factor for the development of cardiovascular diseases in T2DM patients [39]. No literature was found with respect to the rest of the non-collagen peptides in association with GLP-1R agonists. However, the impact of diabetes and/or obesity on these molecules has been reported in the literature. *CD99* transcripts have been stated to increase in abundance in T2DM profiles [40], and Pasello et al. [41] reported involvement of these proteins in biological processes such as cell death and inflammation, including the inflammatory response in the kidneys [42].

*CPSF6* indirectly modulates glucose homeostasis and insulin secretion [43], while *HBA2*, a commonly known marker for anemia and β-thalassaemia, interferes with glycemic markers in T2DM patients [44]. Increased levels of *TTR* have been associated with glucose intolerance, obesity, and decreased pancreatic β-cell percentage in T2DM [45,46]. Increased levels of *TTR* have also been negatively correlated with the glomerular filtration rate, and reduction of retinol-*TTR* levels in CKD patients is considered a potential therapeutic strategy [47]. Along the same lines, increased levels of *SERPINA6* have been identified in obese patients and are reported to play a crucial role in glucose homeostasis, along with reducing insulin resistance and inflammation [48,49]. Benchoula et al. [50] reported that *VGF* is expected to induce obesity while also playing a role in lipolysis and insulin secretion, hence acting as a potential target in T2DM therapy. *CRNN* is reportedly associated with the immune system and acts as an inflammation marker in chronic diseases [51,52]. In addition, elevated levels of *MB*, a known cardiac marker, were reported in T2DM patients [53] and have been associated with insulin resistance, dyslipidemia, abnormal glucose metabolism, and diabetic kidney disease [54]. Similarly, inflammatory mediators have been reported to increase *PIGR* protein levels in the renal tubular cells, linking it to renal injury [55]. Our literature search also indicated that collectively, 15 urinary peptides downregulated by GLP-1R agonists in our study (originating mainly from the collagen protein family), when upregulated, had been previously reported as markers of heart failure [56]. This observation may further indicate the positive effect of GLP-1R agonist treatment.

As a result of this exploratory study and literature search, and based on the observed down-regulation of collagen peptides in the urinary peptidome, we may thus predict for the first time the down-regulation of MMP proteases following treatment with GLP-1R agonists in T2DM patients. In addition, we identified for the first time beneficial down-regulation of the *SERPINA1*, *APOC3*, *CD99*, *CPSF6*, *CRNN*, *SERPINA6*, *HBA2*, *MB*, *VGF*, *PIGR*, and *TTR* urinary peptides by GLP-1R agonist treatment in T2DM patients, which are collectively reported in the literature at elevated levels in T2DM and its associated diseases, playing an important role in increased inflammation, increased insulin secretion, imbalance in glucose homeostasis, and reduced eGFR, amongst others. In line with these results, we propose a plausible molecular mechanism induced by treatment with GLP-1R agonists on the pathophysiology of T2DM (Figure 5). As shown, GLP-1R agonists target three mechanisms involving reduced inflammation, reduced insulin resistance, and the control of glucose homeostasis, which could collectively result in the downregulation of non-collagen proteins (Figure 5). The results of this study may therefore indicate the beneficial effect of GLP-1R agonists in the context of the management of T2DM and the prevention or delaying of the progression of its associated diseases.

Regardless of the novel findings, this study comes with its own limitations. Firstly, the large time interval (4.4 ± 4.11 months) between the pre-treatment and administration of GLP-1R agonists may have resulted in unidentified variations of clinical parameters as well as the composition of urinary peptides, which were not accounted in this study. Secondly, potential administration of multiple anti-hypertensives and GLP-1R agonist drugs at varied dosages and different combinations to the T2DM patients may have produced different effects of the treatment, which were also not analyzed in this study. In addition, within the follow-up we did not observe significant changes in Hb1Ac and BMI. However, it should be emphasized that the study was not powerful enough to detect such changes, which would require about 10 times the number of subjects to be included. In any case, to overcome this short-coming, we additionally, performed a paired Wilcoxon test on a cohort of thirty-two T2DM patients from the same PROVALID study, receiving only the anti-hypertensive drugs and no GLP-1R agonist drugs, that were matched to the GLP-1R agonist treated cohort by age, sex, BMI, SBP, DBP and eGFR. Interestingly, we did not identify any statistically significant urinary peptides between the paired urine samples collected at a similar time interval as in the presented study. Therefore, these observations eliminate the argument that the reported changes in this study could be a result of weight loss, and instead highlight the role of identified proteins in T2DM pathophysiological mechanisms. Furthermore, since only one urinary peptide was identified per non-collagen protein, the reported effects of GLP-1R agonist treatment on these proteins cannot be definitive and require further experimental studies with increased power.

## 4. Materials and Methods

### 4.1. Study Population and Sample Collection

The Prospective Cohort Study in Patients with T2DM for Validation of Biomarkers (PROVALID) study is an observational, prospective cohort study in five European countries, that recruited 4000 patients between the years 2011–2015, aged between 18–75 years, diagnosed with incident or prevalent T2DM (defined as treatment with hypoglycaemic drugs or according to ADA guidelines), irrespective of suffering from chronic kidney disease or not. Only subjects with active malignancy requiring chemotherapy were excluded [57]. The study was approved by the Ethics Committee of the Medical University of Innsbruck (Nr. 1188/2020). Consent was obtained from all patients. Thirty-two T2DM patients recruited within the PROVALID study were included in this study. They received a GLP-1R agonist on top of their standard medication consisting of an inhibitor of the renin angiotensin system at the discretion of their treating physician. No patient was treated with an SGLT2-inhibitor or a mineralocorticoid receptor antagonist (flowchart for the inclusion and exclusion criteria is provided in Figure 6). Urine samples were collected from the patients at their visit just before GLP-1R agonists prescription and labelled as pre-treatment samples. The urine samples collected at the first visit after the treatment initiation were labelled as post-treatment samples. All the samples fulfilling the above criteria (*n* = 64) were stored at −20 °C until peptidomic analysis.

### 4.2. Sample Preparation and CE-MS Analysis

The standard operating protocols describing urine sample preparation and extraction of peptides followed in this study have been applied in numerous studies, as reviewed previously [58]. The CE-MS analysis was conducted as described in detail by Zurbig et al. [59], utilizing a P/ACE MDQ capillary electrophoresis system (Beckman Coulter, Fullerton, CA, USA) coupled with a Micro-TOF II MS (Bruker Daltonic, Bremen, Germany) instrument. Literature evidence on the advantages of CE-MS analysis in terms of reproducibility, sensitivity, precision, and accuracy is extensively available. The relative peptide intensities were normalized based on an internal standard of 29 stable collagen peptides that can be detected naturally in urine samples of healthy and diseased individuals. This calibration was performed to normalize the variability in peptide intensities. The resulting peptides and their normalized intensity values were stored in an internal Microsoft SQL database [60], which enabled the comparison of the pre- and post-treatment urinary peptide profiles. For identification of the peptide sequences, MS/MS-based analysis by a Dionex Ultimate 3000 RSLS nanoflow system (Dionex, Camberley, UK) or a Beckman P/ACE MDQ CE that was coupled to an Orbitrap Velos MS instrument (Thermo Fisher Scientific Inc., Boston, MA, USA) was performed.

### 4.3. Statistical Analysis

All statistical analyses in this study were performed using R programming (R version 4.2.2, R Foundation for Statistical Computing, Vienna, Austria, with IDE: R Studio Version 1.2.5, Boston, MA, USA). As a pre-requisite for the analysis, thresholds of 30% (i.e., ≥ 10 out of 32 datasets per treatment group) peptide frequency were applied. Alongside, area under the receiver operating curve (ROC) curve (AUC) values were calculated by the DeLong approach to compare the urinary peptide profiles between the pre- and post-treated samples; the selected urinary peptides passed a threshold of AUC ≥ 0.60. The normally distributed and continuous datasets generated from the CE-MS-based peptide profiles of the urine samples, obtained from pre-treatment (*n* = 32) and post-treatment (*n* = 32), were compared by a paired Wilcoxon rank-sum test using the row_wilcoxon_paired() function from the matrixTests package version 0.1.9.1. A *p*-value < 0.05 was considered statistically significant and was further adjusted for false discovery rates (FDR) by the Benjamini-Hochberg method [61]. All the plots in this manuscript were created using the ggplot() function from the ggplot2 package version 3.4.1.

### 4.4. Bioinformatic Analysis

Bioinformatics analysis was employed to place the findings of urinary peptidomics within the biological context. Given that changes in peptide levels might be indicative of alterations in protease activity, the proteolytic events responsible for the secretion of the statistically significant urinary peptides were investigated by the open-source online tool “Proteasix” (http://www.proteasix.org, accessed on 6 March 2023). In brief, Proteasix retrieves information from literature and databases like MEROPs, UniProt Knowledgebase (KB), and CutDB. The “Observed Prediction tool” of Proteasix, with search parameters set to default, was utilized in this study. In addition, the relevant parental proteins corresponding to the statistically significant urinary peptides were subjected to network and pathway enrichment analysis utilizing the default settings of the online STRING database [62,63] (http://www.string-db.org, accessed on 15 May 2023). The Kyoto Encyclopedia of Genes and Genomes (KEGG) pathway database, which contains pathway maps representing the current understanding of molecular interaction and reaction networks, was utilized for enrichment analysis. Default settings were used, and pathways with a predicted significance of *p*-value < 0.05 (FDR) were considered statistically significant.

## 5. Conclusions

To conclude, this untargeted peptidomic analysis to identify the effect of GLP-1R agonist treatment on the urinary peptidome of T2DM patients indicated as a prominent finding the downregulation of MMP proteases, as predicted based on the observed collagen peptide changes following GLP-1R agonist treatment. Treatment with these drugs also resulted in a decrease in *SERPINA1*, *APOC3*, *CD99*, *CPSF6*, *CRNN*, *SERPINA6*, *HBA2*, *MB*, *VGF*, *PIGR*, and *TTR* urinary peptides, indicating a potential treatment benefit as many of these proteins are found at increased levels in T2DM patients. The results open the way for larger cohort studies to further characterize the GLP-1R agonist-induced molecular changes and better understand the impact of these drugs on insulin resistance and inflammation in T2DM and its complications.

## Figures and Tables

**Figure 1 ijms-24-13540-f001:**
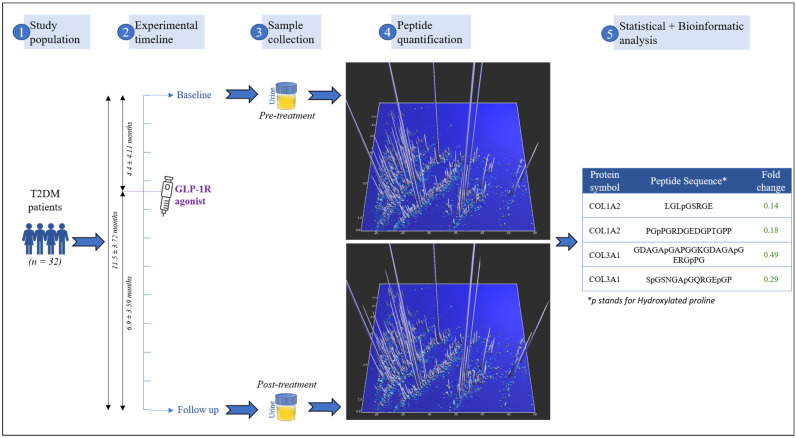
Study design. Urine samples from thirty-two T2DM patients were collected at two time points: pre-treatment and post-treatment with the intervention of GLP-1R agonists at 4.4 ± 4.11 months from the first sample collection. Naturally occurring urinary peptides were quantified in the urine samples by CE-MS analysis, followed by statistical and bioinformatic analysis of the generated urinary peptide profiles.

**Figure 2 ijms-24-13540-f002:**
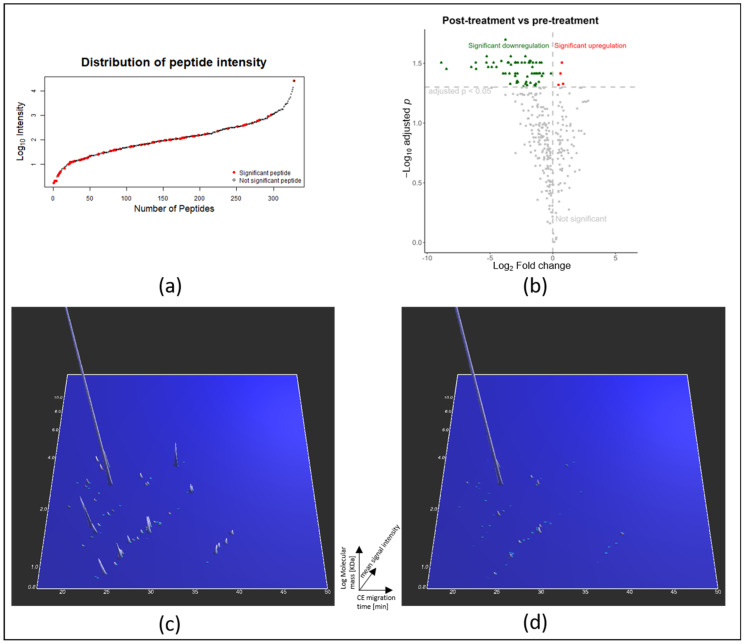
Results of the urinary peptidomic analysis. (**a**) Distribution of peptide intensity for all the 329 sequenced urinary peptides identified in this study; red dots indicate the statistically significant peptides. (**b**) Volcano plot depicting the regulation of the 329 peptides in response to GLP-1R agonist treatment. (*Green dots represent the significantly downregulated peptides; red the significantly upregulated peptides; and gray the non-significant peptides*). (**c**) Urinary CE-MS peptide profiles of the 70 significant peptides during pre-treatment and (**d**) post-treatment.

**Figure 3 ijms-24-13540-f003:**
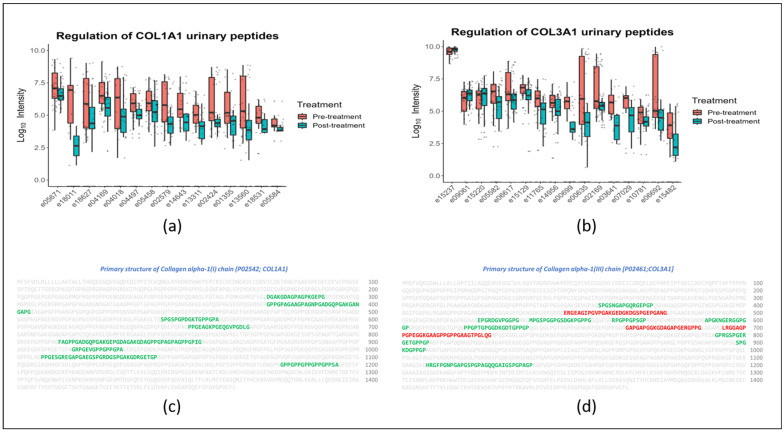
*COL1A1* and *COL3A1* peptides. (**a**) Box and Whisker plots depicting the down-regulation of all the significant *COL1A1* peptides in response to GLP-1R agonist treatment. (**b**) Box and Whisker plots depicting the differential abundance of the *COL3A1* peptides in response to GLP-1R agonist treatment. (**c**) Alignment of the identified peptide sequences in the primary structure of protein *COL1A1*. (**d**) Alignment of the identified peptide sequences in the primary structure of protein *COL3A1*. In (**c**,**d**), *the amino acids in green and red depict the down- and up-regulated peptide sequences, respectively*.

**Figure 4 ijms-24-13540-f004:**
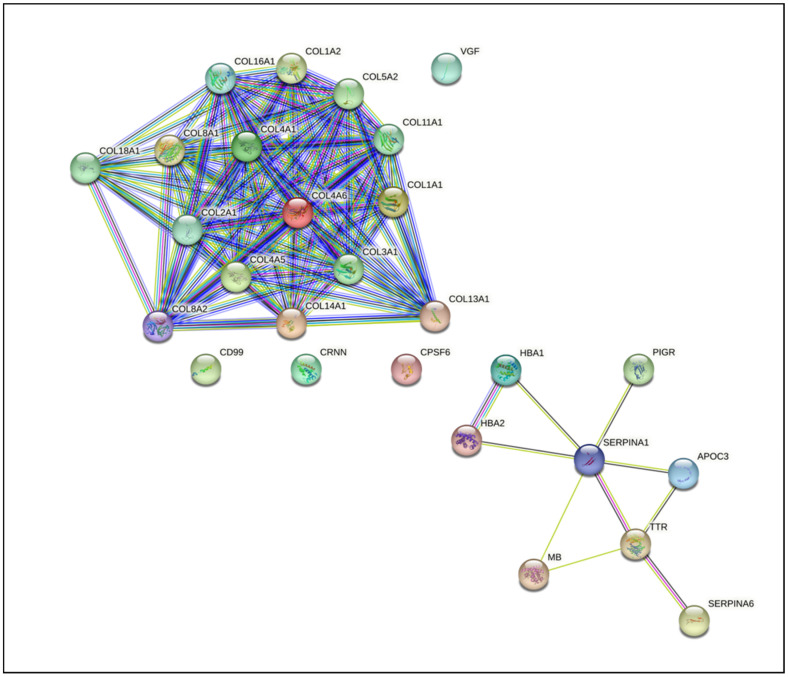
Protein-protein interactome. The network was constructed based on the 26 parental proteins of the 70 GLP-1R agonist-associated urinary peptides.

**Figure 5 ijms-24-13540-f005:**
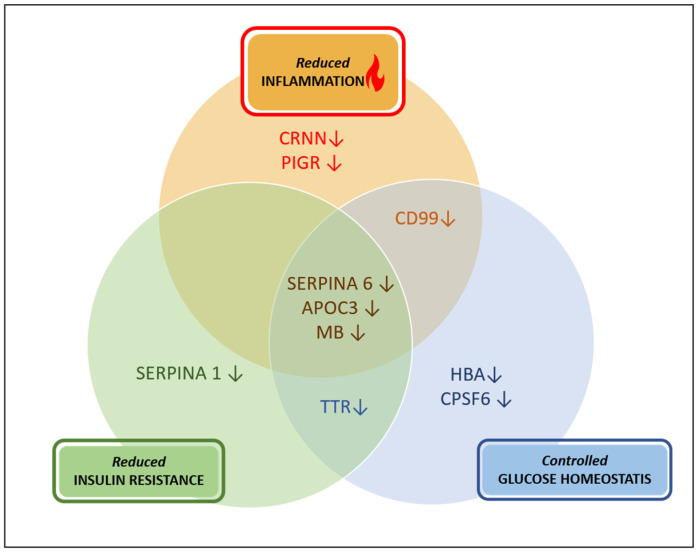
Hypothesis. The beneficial effects of GLP-1R agonist treatment on the different pathophysiological pathways associated with T2DM as suggested by the down-regulated non-collagen peptides, (respective protein names are shown) in each pathway.

**Figure 6 ijms-24-13540-f006:**
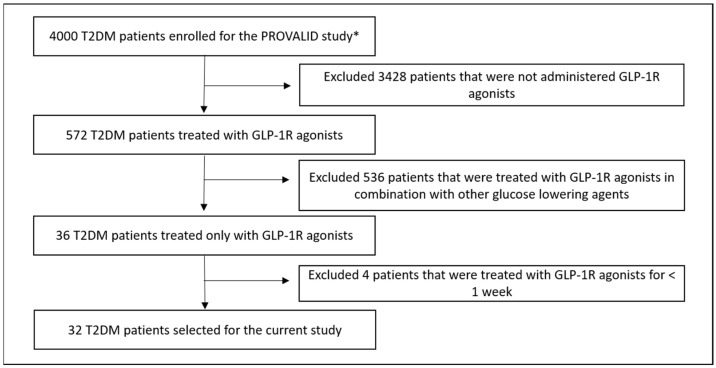
Flowchart summarizing the selection of 32 T2DM patients treated with GLP-1R agonists from the PROVALID study. * PROVALID was an observational study; therefore, patients were treated the way their physicians thought it was appropriate. For this study, we selected individuals that did not receive a GLP-1R agonist at pre-treatment urine sampling and were administered a GLP-1R agonist at post-treatment urine sampling.

**Table 1 ijms-24-13540-t001:** Clinical information of the T2DM patients (mean ± SD are provided) from pre- and post-treatment with GLP-1R agonists.

Clinical Information ^a^	T2DM Patients (*n* = 32)		*p*-Value ^b^
	Pre-Treatment	Post-Treatment	
Age (years)	63.7 ± 7.3		
Sex (% Females)	56.3		
HbA1C (%)	8.0 ± 1.4	8.0 ± 1.5	0.731
BW (kg)	98.3 ± 16.1	98.8 ± 20.3	0.771
BMI (kg/m^2^)	33.2 ± 6.3	33.2 ± 6.9	0.915
SBP (mmHg)	139.2 ± 14.4	137.1 ± 12.6	0.385
DBP (mmHg)	81.1 ± 7.7	81.3 ± 8.2	0.906
eGFR (mL/min/1.73 m^2^)	67.7 ± 14.5	67.6 ± 15.6	0.965
UACR (mg/g)	33.6 ± 58.6	25.5 ± 44.8	0.248
UCREA (mg/dL)	103.9 ± 41.6	96.6 ± 50.7	0.429

^a^ HbA1C: Hemoglobin A1C; BW: Body weight; BMI: Body mass index; SBP: Systolic blood pressure; DBP: Diastolic blood pressure; eGFR: estimated glomerular filtration rate; UACR: Albuminuria to Creatinine ratio; UCREA: Urinary Creatinine; ^b^ *p*-values were obtained from paired Wilcoxon rank sum (continuous variables) and Chi-squared test (categorical variable).

**Table 2 ijms-24-13540-t002:** A list of 26 proteins yielding the 70 statistically significant urinary peptides in response to GLP-1R agonist treatment (↓ refers to down regulation and ↑ refers to upregulation following treatment).

UniProt ID	Gene Symbol	Protein Name	Number of Peptides	
			Total	Regulation after Treatment
P02461	*COL3A1*	Collagen alpha-1(III) chain	16	↓ (13) + ↑ (3)
P02452	*COL1A1*	Collagen alpha-1(I) chain	15	↓
P08123	*COL1A2*	Collagen alpha-2(I) chain	10	↓ (9) + ↑ (1)
P02458	*COL2A1*	Collagen alpha-1(II) chain	3	↓
P02462	*COL4A1*	Collagen alpha-1(IV) chain	2	↓
P05997	*COL5A2*	Collagen alpha-2(V) chain	2	↓
P12107	*COL11A1*	Collagen alpha-1(XI) chain	2	↓
P27658	*COL8A1*	Collagen alpha-1(VIII) chain	1	↓
Q5TAT6	*COL13A1*	Collagen alpha-1(XIII) chain	1	↓
Q05707	*COL14A1*	Collagen alpha-1(XIV) chain	1	↓
Q07092	*COL16A1*	Collagen alpha-1(XVI) chain	1	↓
P39060	*COL18A1*	Collagen alpha-1(XVIII) chain	1	↓
P25067	*COL8A2*	Collagen alpha-2(VIII) chain	1	↓
P29400	*COL4A5*	Collagen alpha-5(IV) chain	1	↓
Q14031	*COL4A6*	Collagen alpha-6(IV) chain	1	↓
P01009	*SERPINA1*	Alpha-1-antitrypsin	1	↓
P02656	*APOC3*	Apolipoprotein C-III	1	↓
P14209	*CD99*	CD99 antigen	1	↓
Q16630	*CPSF6*	Cleavage and polyadenylation specificity factor subunit 6	1	↓
Q9UBG3	*CRNN*	Cornulin	1	↓
P08185	*SERPINA6*	Corticosteroid-binding globulin	1	↓
P69905	*HBA1*; *HBA2*	Hemoglobin subunit alpha	1	↓
P02144	*MB*	Myoglobin	1	↓
O15240	*VGF*	Neurosecretory protein VGF	1	↓
P01833	*PIGR*	Polymeric immunoglobulin receptor	1	↓
P02766	*TTR*	Transthyretin	1	↓

## Data Availability

Data can be made available upon request directed to the corresponding author. Proposals will be reviewed and approved by the investigators and collaborators based on scientific merit. After approval of a proposal, data will be shared through a secure online platform.

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
