# Peer review of "Exploratory Study Analyzing the Urinary Peptidome of T2DM Patients Suggests Changes in ECM but Also Inflammatory and Metabolic Pathways Following GLP-1R Agonist Treatment"

_ijms, 2023, doi:10.3390/ijms241713540_

Round 1

Reviewer 1 Report

The authors have performed a study about the impact of GLP-1 agonists on urine peptidome. The study demonstrated novel findings and it contributes to the current knowledge about GLP-1 agonists.

1. Introduction: The authors are kindly requested to reduce the amount of info about T2DM and focus mainly on the urine peptidome, as it seems to contribute a lot in T2DM-related chronic kidney diseases. Also, they are kindly requested to elucidate more on the aims at the last paragraph of Introduction.

2. The authors are kindly requested to adjust their manuscript according to the appropriate statement of the EQUATOR Network.

3. The sample size of this study is small. There is no flowchart, nor dates of recruitment, inclusion or exclusion criteria, whatsoever. The authors are kindly requested to add all the missing info.

4. Discussion needs more work. The authors need to emphasise their findings. Are their findings novel? What new do they contribute?

Author Response

“The authors have performed a study about the impact of GLP-1 agonists on urine peptidome. The study demonstrated novel findings and it contributes to the current knowledge about GLP-1 agonists.”

  • Thank you very much for this positive comment and acknowledging the efforts made behind the manuscript.

“1. Introduction: The authors are kindly requested to reduce the amount of info about T2DM and focus mainly on the urine peptidome, as it seems to contribute a lot in T2DM-related chronic kidney diseases. Also, they are kindly requested to elucidate more on the aims at the last paragraph of Introduction.”

  • Thank you very much for this critical and helpful comment.
  • We agree with you and have made major changes in the Introduction section, which include reducing information on T2DM and its treatment (Line 43, 48, 60 amongst others), and including information on research studies applying urinary peptidomics in the field of T2DM and CKD, as was kindly suggested by you (Line 72).
  • We also elucidated on aims in the last paragraph to “In this exploratory study, we aimed to assess the effect of treatment with GLP-1R agonists on the urinary peptidome of T2DM patients, in an untargeted peptidomics approach, aiming to ultimately shed some light on the underlying molecular mechanisms defining response.” (Line 81)

“2. The authors are kindly requested to adjust their manuscript according to the appropriate statement of the EQUATOR Network.”

  • Thank you for this meaningful suggestion. The manuscript has now been revised based on the Strengthening the Reporting of Observational Studies in Epidemiology (STROBE) Statement of the EQUATOR network, under the category of Cohort study.
  • As required, the manuscript now includes information on setting, locations, and relevant dates, including periods of recruitment, exposure, follow-up, and data collection for the 32 patients included in this study. (Line 324, Materials and Methods section 4.1 Study population and sample collection)

“3. The sample size of this study is small. There is no flowchart, nor dates of recruitment, inclusion, or exclusion criteria, whatsoever. The authors are kindly requested to add all the missing info.”

  • This is a well taken point and we are sorry for missing out on this crucial information.
  • The Materials and Methods section 4.1 Study population and sample collection has been altered accordingly to “The Prospective Cohort Study in Patients with T2DM for Validation of Biomarkers (PROVALID) study is an observational, prospective cohort study in five European countries, that recruited 4000 patients between the years 2011-2015, aged between 18-75 years, diagnosed with incident or prevalent T2DM (defined as treatment with hypoglycaemic drugs or according to ADA guidelines), irrespective of suffering from chronic kidney disease or not. Only subjects with active malignancy requiring chemotherapy were excluded [57]. The study was approved by the Ethics Committee of the Medical University of Innsbruck (Nr. 1188/2020). Consent was obtained from all patients. Thirty-two T2DM patients recruited within the PROVALID study were included in this study. They received a GLP-1R agonist on top of their standard medication consisting of an inhibitor of the renin angiotensin system at the discretion of their treating physician. No patient was treated with an SGLT2-inhibitor or a mineralocorticoid receptor antagonist (flowchart for the inclusion and exclusion criteria is provided in Figure 6). Urine samples were collected from the patients at their visit just before GLP-1R agonists prescription and labelled as pre-treatment samples. The urine samples collected at the first visit after the treatment initiation were labelled as post-treatment samples. All the samples fulfilling the above criteria (n=sixty-four) were stored at -20°C until peptidomic analysis.” (Line 324)
  • We have furthermore included a workflow representing the inclusion and exclusion criteria, as depicted in Figure 6.
  • Again, we apologize for missing out on this crucial information.

“4. Discussion needs more work. The authors need to emphasise their findings. Are their findings novel? What new do they contribute?”

  • This was a very interesting observation, and a well taken-point. We have made many changes to the Discussion section explicitly emphasizing the novel findings and providing references to already published findings supporting our study. (Line 250, 252, 258, 265)
  • We have also expanded furthermore on the hypothesis presented in this study, as a result of the findings of this study and literature search performed. (Line 282)

Reviewer 2 Report

Dear Editor and Dear Authors,

This is an interesting study evaluating the systemic effects of GLP-1RAs on urine peptidome. The paper may provide additional information about the systemic and local (renal) impact of GLP-1RAs in T2D. Nevertheless, a relevant adjustment is recommended to improve the quality of the manuscript and the soundness of the results. 

1. Title. The title should be informative on the study design and/or results.

2. Methods. Please consider including either a flow chart or a brief section to clarify the selection of the study participants better.

3. Methods. It's just a comment, but why did you consider intensifying the treatment by adding a GLP-1RA to background therapy after a mean of 4 months? Please also clarify the background treatment (were patients on metformin?) and the criteria you consider patients for GLP-1RA intensification. 

4. Methods. Understanding why the urinary peptidomic analysis resulted in a list of 329 sequenced peptides could be a little unclear. Please provide a more detailed explanation of the method and cite specific references, if any.

5. Discussion. Could the down-regulation of most collagen-related peptides be attributable to a GLP-1RA-induced down-regulation of collagen synthesis? This could be an issue, especially for an underestimated chronic comorbidity/complication of T2D (NASH).

6. Discussion. Even if some hypotheses could be purely speculative, the manuscript could be improved by explaining if the down-regulation of these peptides (some of which probably have pro-inflammatory and pro-fibrotic properties) may have a positive effect on glomerular and tubular structures, ultimately posing the bases for a better understanding of renal benefits of GLP-1RAs.

Please check the text for typing errors and minor mistakes.

Author Response

“Dear Editor and Dear Authors,

This is an interesting study evaluating the systemic effects of GLP-1RAs on urine peptidome. The paper may provide additional information about the systemic and local (renal) impact of GLP-1RAs in T2D. Nevertheless, a relevant adjustment is recommended to improve the quality of the manuscript and the soundness of the results.”

  • We cordially thank the reviewer for this supportive comment. We have made significant changes to the manuscript to improve the quality as suggested.

“1. Title. The title should be informative on the study design and/or results.”

  • This was a very interesting observation, for which we are grateful. We have changed the title of the manuscript to “Exploratory study analysing the urinary peptidome of T2DM patients suggest changes in ECM following GLP-1R agonists treatment”, which we believe better describes the aim and findings of the manuscript.

“2. Methods. Please consider including either a flow chart or a brief section to clarify the selection of the study participants better.”

  • We would like to apologize for missing information on the inclusion and exclusion criteria and have revised the Materials and Methods section 4.1 Study population and sample collection has been altered accordingly to “The Prospective Cohort Study in Patients with T2DM for Validation of Biomarkers (PROVALID) study is an observational, prospective cohort study in five European countries, that recruited 4000 patients between the years 2011-2015, aged between 18-75 years, diagnosed with incident or prevalent T2DM (defined as treatment with hypoglycaemic drugs or according to ADA guidelines), irrespective of suffering from chronic kidney disease or not. Only subjects with active malignancy requiring chemotherapy were excluded [57]. The study was approved by the Ethics Committee of the Medical University of Innsbruck (Nr. 1188/2020). Consent was obtained from all patients. Thirty-two T2DM patients recruited within the PROVALID study were included in this study. They received a GLP-1R agonist on top of their standard medication consisting of an inhibitor of the renin angiotensin system at the discretion of their treating physician. No patient was treated with an SGLT2-inhibitor or a mineralocorticoid receptor antagonist (flowchart for the inclusion and exclusion criteria is provided in Figure 6). Urine samples were collected from the patients at their visit just before GLP-1R agonists prescription and labelled as pre-treatment samples. The urine samples collected at the first visit after the treatment initiation were labelled as post-treatment samples. All the samples fulfilling the above criteria (n=sixty-four) were stored at -20°C until peptidomic analysis.” (Line 324)
  • We have furthermore included a workflow representing the inclusion and exclusion criteria, as depicted in Figure 6.
  • Again, we apologize for missing out on this crucial information.

“3. Methods. It's just a comment, but why did you consider intensifying the treatment by adding a GLP-1RA to background therapy after a mean of 4 months? Please also clarify the background treatment (were patients on metformin?) and the criteria you consider patients for GLP-1RA intensification.”

  • Thank you for your comment and we apologise if the information was not clear enough.
  • PROVALID was an observational study, so patients were treated the way their physicians thought it was appropriate. For DC-ren, we selected individuals based on the fact that they did not have a GLP-1R agonist at one time of sampling and had one the next visit after one year. So, the 4 months were driven by patient selection but not by a prospective study protocol demand. 
  • We hope this explanation answers your question. We have also included this in the flowchart figure with an asterisk. (Figure 6 caption)

“4. Methods. Understanding why the urinary peptidomic analysis resulted in a list of 329 sequenced peptides could be a little unclear. Please provide a more detailed explanation of the method and cite specific references, if any.”

  • We would like to kindly clarify that the explanation was provided in the Materials and Methods 4.1 section 4.3 Statistical analysis
  • Nevertheless, we appreciate that the Results section was not clearer and have included the text “This exploratory urinary peptidomic analysis resulted in a list of 329 sequenced peptides that passed the thresholds of frequency (i.e., being detected in ≥ 10 out of 32 datasets or with at least 30% frequency in one group) and area under the receiver operating curve (AUC ≥ 0.60), as described in the methods section.” (Line 116)

“5. Discussion. Could the down-regulation of most collagen-related peptides be attributable to a GLP-1RA-induced down-regulation of collagen synthesis? This could be an issue, especially for an underestimated chronic comorbidity/complication of T2D (NASH).”

  • We would like to thank the reviewer for this well taken comment. We have included the text “The observed downregulation of the collagen peptides in our study could therefore represent attenuated degradation of the mature collagen due to increased resistance to proteolytic cleavage or expression of protease inhibition as earlier suggested [23,28], nevertheless, the possibility of decreased collagen synthesis as a result of treatment cannot also be ruled out.” (Line 221)

“6. Discussion. Even if some hypotheses could be purely speculative, the manuscript could be improved by explaining if the down-regulation of these peptides (some of which probably have pro-inflammatory and pro-fibrotic properties) may have a positive effect on glomerular and tubular structures, ultimately posing the bases for a better understanding of renal benefits of GLP-1RAs.”

  • We would like to thank you for this very helpful comment.
  • We revised the Discussion section including information on the beneficial effect of treatment with GLP-1R agonists in the downregulation of identified urinary peptides in T2DM and its associated diseases like cardiovascular and chronic kidney diseases. (Line 250, 252, 258, 265)
  • We have tried to include an appraisal of relevant publications supporting the beneficial effect observed, providing also a summary of the more reproducible findings (in our opinion), as well as expanded the Discussion section. (Line 282)